# Differential neutralization and inhibition of SARS-CoV-2 variants by antibodies elicited by COVID-19 mRNA vaccines

Li Wang[1,8], Markus H. Kainulainen [1,8], Nannan Jiang [1,8], Han Di[1,8], Gaston Bonenfant [1,8], Lisa Mills[1], Michael Currier[1], Punya Shrivastava-Ranjan[1], Brenda M. Calderon[1], Mili Sheth[1], Brian R. Mann[1], Jaber Hossain[1], Xudong Lin[1], Sandra Lester[1], Elizabeth A. Pusch [1], Joyce Jones[1], Dan Cui[1], Payel Chatterjee[1], M. Harley Jenks[1], Esther K. Morantz[1], Gloria P. Larson [1], Masato Hatta[1], Jennifer L. Harcourt[1], Azaibi Tamin[1], Yan Li[1], Ying Tao[1], Kun Zhao[1], Kristine Lacek [1], Ashley Burroughs[1], Wei Wang [1], Malania Wilson[1], Terianne Wong[1], So Hee Park[1], Suxiang Tong[1], John R. Barnes[1], Mark W. Tenforde [1], Wesley H. Self[2], Nathan I. Shapiro[3], Matthew C. Exline[4], D. Clark Files[5], Kevin W. Gibbs [5], David N. Hager[6], Manish Patel[1], Alison L. Halpin[1], Laura K. McMullan [1], Justin S. Lee[1], Hongjie Xia [7], Xuping Xie [7], Pei-Yong Shi [7], C. Todd Davis[1], Christina F. Spiropoulou[1], Natalie J. Thornburg [1], M. Steven Oberste[1], Vivien G. Dugan[1], SSEV Bioinformatics Working Group*, David E. Wentworth [1✉] & Bin Zhou [1✉]

The evolution of severe acute respiratory syndrome coronavirus 2 (SARS-CoV-2) has resulted in the emergence of new variant lineages that have exacerbated the COVID-19 pandemic. Some of those variants were designated as variants of concern/interest (VOC/ VOI) by national or international authorities based on many factors including their potential impact on vaccine-mediated protection from disease. To ascertain and rank the risk of VOCs and VOIs, we analyze the ability of 14 variants (614G, Alpha, Beta, Gamma, Delta, Epsilon, Zeta, Eta, Theta, Iota, Kappa, Lambda, Mu, and Omicron) to escape from mRNA vaccine-induced antibodies. The variants show differential reductions in neutralization and replication by post-vaccination sera. Although the Omicron variant (BA.1, BA.1.1, and BA.2) shows the most escape from neutralization, sera collected after a third dose of vaccine (booster sera) retain moderate neutralizing activity against that variant. Therefore, vaccination remains an effective strategy during the COVID-19 pandemic.

[1] COVID-19 Emergency Response, Centers for Disease Control and Prevention, Atlanta, GA, USA. [2] Vanderbilt University, Nashville, TN, USA. [3] Harvard University, Cambridge, MA, USA. [4] Ohio State University, Columbus, OH, USA. [5] Wake Forest Baptist Medical Center, Winston-Salem, NC, USA. [6] Johns Hopkins University, Baltimore, MD, USA. [7] University of Texas Medical Branch, Galveston, TX, USA. [8] These authors contributed equally: Li Wang, Markus H. Kainulainen, Nannan Jiang, Han Di, Gaston Bonenfant. *A list of authors and their affiliations appears at the end of the paper. ✉email: dwentworth@cdc.gov; bzhou@cdc.gov

SARS-CoV-2 was first detected in China in December 2019[1]; within four months, a variant with a D614G substitution in the viral spike protein became the predominant circulating strain globally[2]. While the D614G variant did not evade antibody-mediated neutralization, enhanced replication and transmissibility of the variant were confirmed in multiple animal models by different groups[3–5]. Enhanced transmissibility and a larger infected population likely led to diversification of the D614G variant into many new lineages. In December 2020, the United Kingdom reported increased transmission of a novel variant of concern (VOC) 202012/01[6], also referred to as the Alpha (or B.1.1.7, Pango nomenclature) variant[7]. The Alpha variant rapidly disseminated and became the predominant circulating strain in many countries, including the United States (US)[8,9] (Fig. 1). Meanwhile, the Beta (i.e., B.1.351) and Gamma (i.e., P.1) variants were first detected in South Africa in May 2020 and in Brazil in November 2020, respectively, where each variant became the predominant lineage in its respective geographic region[10–12]. By August 2021, the Delta variant (B.1.617.2), which was first identified in India[13], had displaced the Alpha variant and become the predominant variant within the US (Fig. 1) and globally. The Omicron variant (B.1.1.529), first reported in late November 2021, has spread rapidly and displaced the Delta variant (Fig. 1). The World Health Organization (WHO) and national health authorities, such as the US government SARS-CoV-2 Interagency Group (US-SIG), have designated selected SARS-CoV-2 variants as VOCs or VOIs (Supplementary Table 1) based on genomic analysis, transmissibility, disease severity, and, most importantly, impact on the performance of therapeutics or vaccines. Continuous monitoring and rapid characterization of VOCs, VOIs, and other new variants are critical to alleviating the devastating impact of the current pandemic.

In this work, we systematically evaluate the neutralization efficiency of U.S. mRNA vaccinee sera against representative viruses of all current and past VOCs and VOIs designated by the WHO Virus Evolution Working Group. We also examine the inhibitory effect of post-second dose and post-third dose sera on the replication of Omicron and other variants in cell culture.

## Results

**Generation of recombinant SARS-CoV-2 reporter viruses.** To characterize emerging variants in the shortest time frame, particularly in periods which lacked clinical isolates in the US, we generated SARS-CoV-2 fluorescent reporter viruses with VOC and VOI spike mutations in the progenitor Wuhan-Hu-1 virus (designated as 614D in this study) by reverse genetics (Supplementary Table 1). The reporter SARS-CoV-2 viruses were designed to behave similarly to their clinical isolate counterparts in neutralization assays due to an identical variant spike protein, which is the sole antigen of all vaccines authorized in the US. The spike glycoproteins in different strains of a VOC/VOI have subtle differences, and the sequence of the spike protein used in our studies represent the consensus of that VOC/VOI or represent a more divergent one from the consensus of that VOC/VOI (Supplementary Table 1). For example, the spike of the Beta variant tested includes R246I in the N-terminal domain (NTD), which is not found in all Beta lineage viruses. The recombinant reporter virus was generated by electroporation of in vitro transcribed viral genomic RNA into VeroE6 cells expressing SARS-CoV-2 nucleocapsid protein (VeroE6-N cells). The resultant virus was propagated in VeroE6-TMPRSS2 cells[14] to make a working stock, which was sequenced to confirm the presence of designed spike mutations and absence of unwanted mutations. The focus forming units (FFU) of the working stock was determined by counting the fluorescent foci on a monolayer of VeroE6-TMPRSS2 cells.

**Neutralizing activity of post-second dose vaccinee sera.** After the Pfizer-BioNTech mRNA vaccine, BNT162b2, and the Moderna mRNA vaccine, mRNA-1273, received Emergency Use Authorization from the U.S. Food and Drug Administration and were recommended by the CDC and administered in eligible populations, we collected sera from volunteer vaccinees 2–6 weeks after they received the second dose of vaccine (i.e., completion of primary series) and those sera are referred specifically in this study as post-second dose sera. Using a focus reduction neutralization test (FRNT), we detected minimal impact of the D614G substitution (virus designated as 614G) on the neutralizing activity of those sera, compared to the progenitor

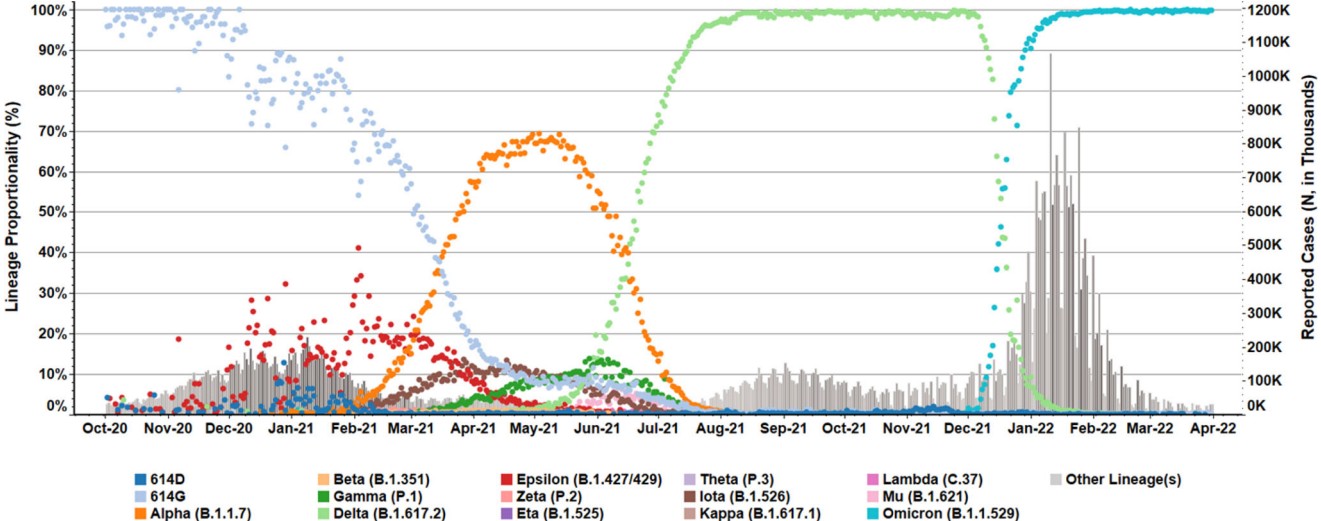

**Fig. 1 Prevalence of SARS-CoV-2 variants in the United States.** SARS-CoV-2 variant prevalence is highlighted for US clinical specimens processed within the National SARS-CoV-2 Strain Surveillance network, the CDC-contracted diagnostic and research laboratories, and the US baseline surveillance program. Daily incidence (dot icons, 0 to 100%) from October 2020 to March 2022 were indicated for key WHO labeled variants (Pangolin lineage in parentheses), and specimens encoding critical sequence markers (614D/G) but do not belong to those variants. Daily, reported clinical cases are summarized in the bar graph (right-side, dual Y-axis).

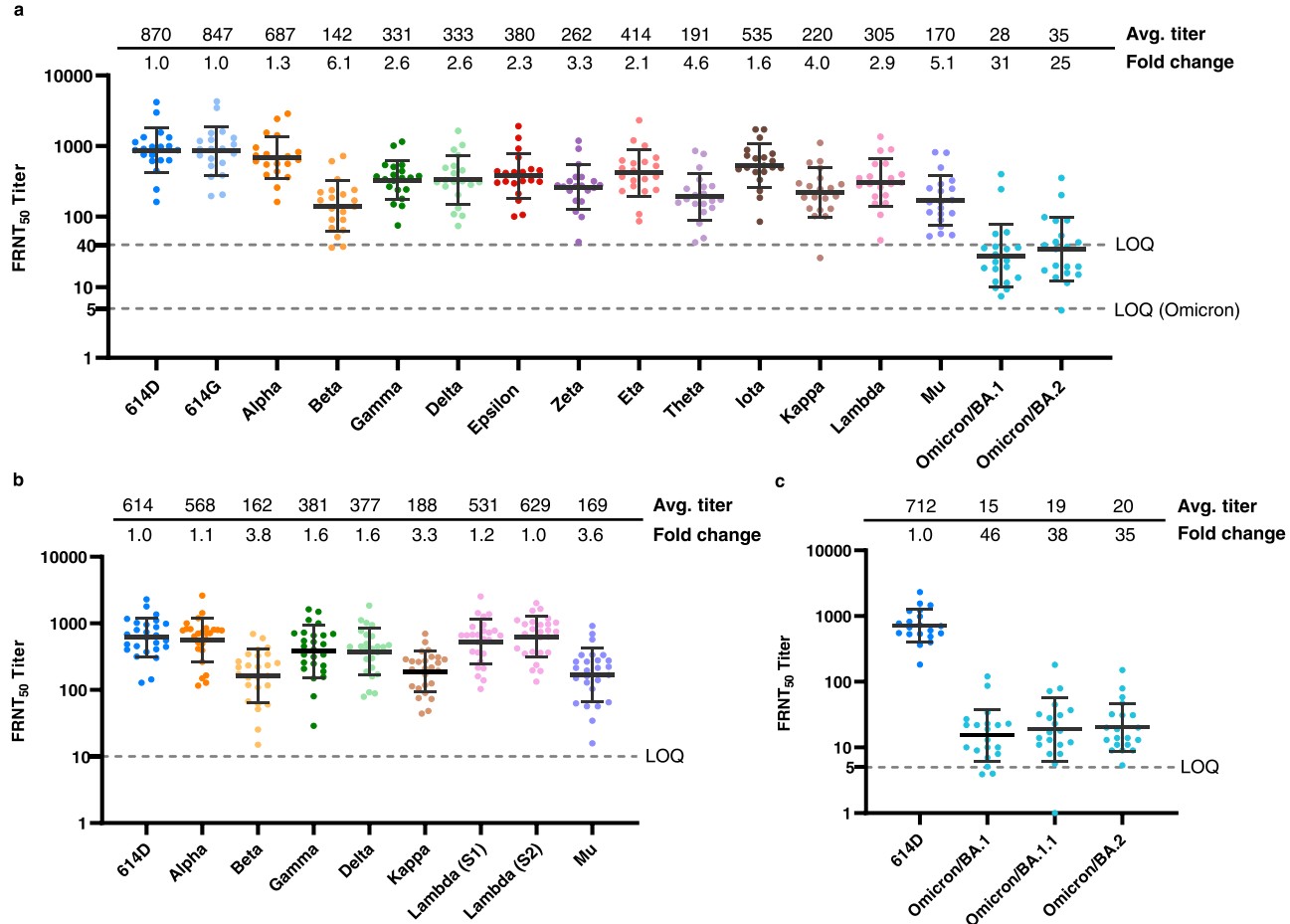

**Fig. 2 Neutralizing activity of mRNA vaccinee sera (post-second dose) against live SARS-CoV-2 viruses.** Each dot represents the neutralizing titer (FRNT$_{50}$) of an individual serum sample against a specific SARS-CoV-2 virus, which is labeled on the X-axis. Different colors represent the progenitor virus or different variants (specific mutations provided in Supplementary Table 1). Vaccinee serum samples were obtained from people who were vaccinated 2–6 weeks post second dose. The geometric mean FRNT$_{50}$ titers are represented in the graph as bars on top of the dots with geometric standard deviation. The geometric mean FRNT$_{50}$ titers (avg. titer) and average fold changes relative to reference virus 614D (set as 1-fold) are shown on the top of the graph. For each variant, the average fold change is the geometric mean of the individual FRNT$_{50}$ ratios (614D/variant) calculated for each serum sample. Dashed line represents the limit of quantitation (LOQ). For statistical analysis, a two-tailed Wilcoxon matched-pairs signed-rank test was performed by comparing each variant with 614D. Test statistics and P value are summarized in Supplementary Table 2. Source data are provided as a Source Data file. **a** Representative reporter viruses of all past and current WHO designated SARS-CoV-2 VOCs and VOIs were tested. N = 20 biologically independent sera examined over 16 viruses. LOQ = 5 for Omicron and LOQ = 40 for all other viruses. The average fold changes of all viruses differ significantly (P < 0.0001) from 614D, except for 614G (P = 0.8124). **b** VOCs and selected VOIs isolated from clinical specimens were tested. N = 26 biologically independent sera examined over 9 viruses. The average fold changes of all viruses differ significantly (P < 0.0001) from 614D, except for Alpha (P = 0.4833) and the two Lambda viruses (Lambda S1 and S2, P = 0.0796 and 0.7265, respectively). **c** Omicron subvariant BA.1, BA.1.1, and BA.2 isolated from clinical specimens were tested. N = 20 biologically independent sera examined over four viruses. The average fold changes of all viruses differ significantly (P < 0.0001) from 614D.

614D reference virus, of which the spike sequence is most closely related to the sequence used in mRNA vaccines (Fig. 2a). The Alpha variant (B.1.1.7) showed slightly decreased neutralizing antibody titers while the Gamma (P.1), Delta (B.1.617.2), Epsilon (B.1.427/B.1.429), Zeta (P.2), Eta (B.1.525), Iota (B.1.526), and Lambda (C.37) variants showed greater titer reductions but were <4-fold reduced compared to the 614D reference virus. The Beta (B.1.351), Theta (P.3), Kappa (B.1.617.1), and Mu (B.1.621) variants showed 4–6-fold reductions in titers. The Omicron (B.1.1.529) variant displayed the greatest escape from neutralization with a 31-fold (BA.1) and 25-fold (BA.2) reduction compared to 614D (Fig. 2a).

As the prevalence of variants rose within the US (Fig. 1), the Centers for Disease Control and Prevention (CDC) received an increasing number of clinical specimens from state public health

laboratories and other CDC collaborating laboratories through the National SARS-CoV-2 Strain Surveillance (NS3) system (https://www.cdc.gov/coronavirus/2019-ncov/variants/cdc-role-surveillance.html). We isolated representative variants from these clinical specimens for characterization by FRNT and sequenced the stocks to ensure the spike correctly represented the appropriate variant lineage. Although the neutralizing titers differed slightly, the fold reductions compared to the 614D reference virus were generally consistent between the reporter viruses and clinical isolates (Fig. 2). Prior to the emergence of Omicron, the Beta variant was the most resistant to neutralization, followed by Mu and Kappa. The Gamma and Delta variants showed modest escape from neutralization, and the Alpha variant neutralization was not significantly reduced as compared to the 614D reference virus (Fig. 2b). Lambda was the only variant that

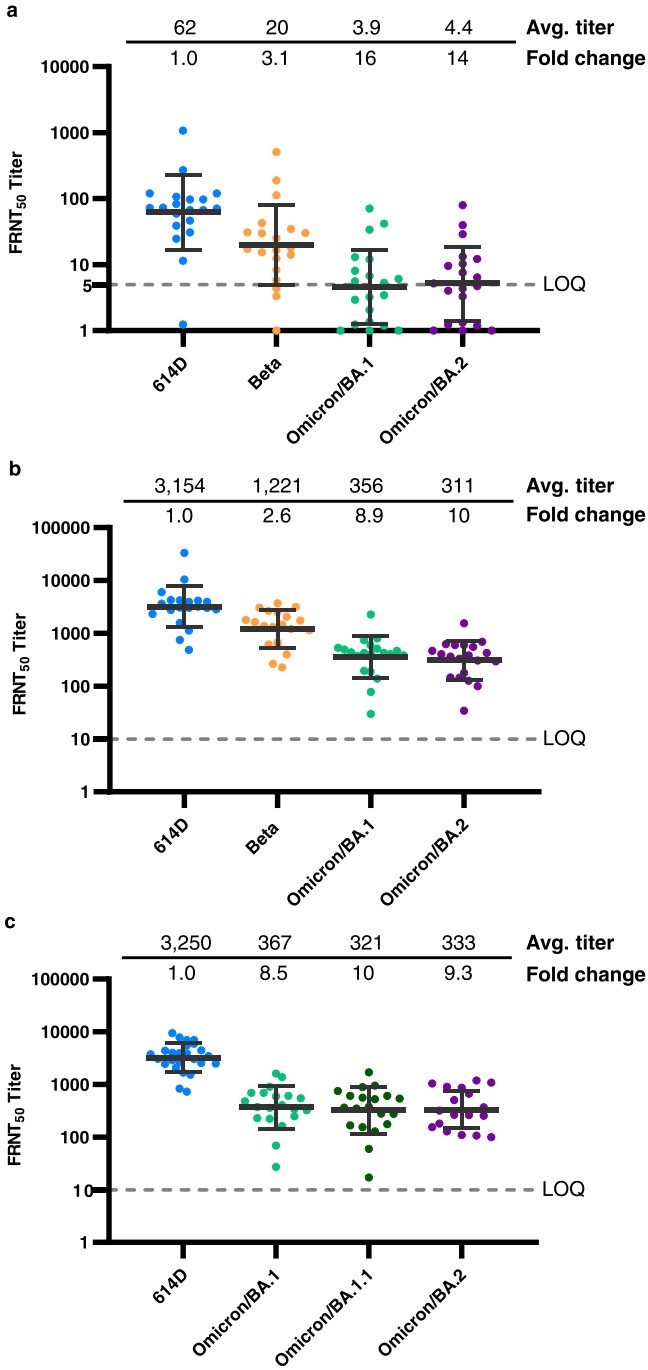

**Fig. 3 Neutralizing activity of mRNA vaccinee sera (pre- and post-booster) against live SARS-CoV-2 viruses.** Each dot represents the neutralizing titer ($FRNT_{50}$) of an individual serum sample against a specific SARS-CoV-2 virus, which is labeled on the X-axis. Different colors represent the progenitor virus or different variants and subvariants. The geometric mean $FRNT_{50}$ titers are represented in the graph as bars on top of the dots with geometric standard deviation. The geometric mean $FRNT_{50}$ titers (avg. titer) and average fold changes relative to reference virus 614D (set as 1-fold) are shown on the top of the graph. Dashed line represents the limit of quantitation (LOQ). For statistical analysis, a two-tailed Wilcoxon matched-pairs signed-rank test was performed by comparing each variant with 614D. Test statistics and $P$ value are summarized in Supplementary Table 2. Source data are provided as a Source Data file. **a** 614D, Beta, and Omicron subvariants BA.1 and BA.2 reporter viruses were tested. $N = 20$ biologically independent pre-booster sera (6–7 months post-second dose). The average fold changes of all variants differ significantly ($P < 0.0001$) from 614D. LOQ = 5. **b** 614D, Beta and Omicron subvariants BA.1 and BA.2 reporter viruses were tested. $N = 20$ biologically independent post-booster sera (2–6 weeks post-third dose). The average fold changes of all viruses differ significantly ($P < 0.0001$) from 614D. LOQ = 10. **c** 614D and Omicron subvariants BA.1, BA.1.1, and BA.2 isolated from clinical specimens were tested. $N = 20$–25 biologically independent post-booster sera examined over four viruses. The average fold changes of all viruses differ significantly ($P < 0.0001$) from 614D. LOQ = 10.

After the first case of the Omicron variant was confirmed in the US on December 1 of 2021, we isolated multiple Omicron variants from clinical specimens and determined their susceptibility to the post-second dose sera. Largely consistent with the results from recombinant reporter viruses (Fig. 2a), the Omicron BA.1 and BA.2 isolates showed a 46-fold and 35-fold reduction compared to the 614D reference virus, respectively (Fig. 2c). In addition, an Omicron BA.1.1 variant showed a 38-fold reduction compared to 614D (Fig. 2c). These results underscored the striking difference between the Omicron variants and the earlier variants in their ability to escape from neutralization by post-second dose mRNA vaccine elicited antibodies.

**Neutralizing activity of post-third dose vaccinee sera**. As the third dose of COVID-19 mRNA vaccine (booster) was authorized, recommended and administered in eligible US populations, we collected pre-booster sera from vaccinees on the day of booster vaccination (6 – 7 months after receiving the second dose of mRNA vaccine) and collected post-booster sera 2–6 weeks after that. The neutralizing activity against the homologous 614D reporter virus decreased by 14-fold from 870 $FRNT_{50}$ in the post-second-dose sera, which in this study refers specifically to the sera collected 2–6 weeks after receiving the second dose of mRNA vaccine, to 62 $FRNT_{50}$ in the pre-booster sera (Figs. 2a and 3a). Importantly, the neutralizing activity against the Omicron BA.1 and BA.2 subvariants decreased by 7–8-fold from 28–35 $FRNT_{50}$ in the post-second dose sera to about 4 $FRNT_{50}$ in the pre-booster sera, which indicates the neutralizing antibody protection against Omicron was minimal 6–7 months after receiving the second dose of mRNA vaccine (Fig. 3a). In contrast, the third dose of vaccine effectively boosted the neutralizing titer against the 614D virus by 51-fold to 3154 $FRNT_{50}$ and against the Omicron BA.1 and BA.2 subvariants by 71–91-fold to 311–356 $FRNT_{50}$ (Fig. 3b). The post-third dose sera were also assessed against three Omicron subvariants (BA.1, BA.1.1, and BA.2) isolated from clinical specimens (Fig. 3c). The neutralizing titers were very close between the reporter viruses and the clinical isolates and among the three Omicron BA.1, BA.1.1, and BA.2 subvariants, which were the

showed a difference between the reporter virus and the clinical isolate, which had an approximate 3-fold reduction versus 1–1.2-fold reduction in neutralizing titers, respectively (Fig. 2a, b). The Lambda variant has at least 14 substitutions/deletions in the spike protein (Supplementary Table 1) and the limited resistance to neutralization (1–1.2-fold) of clinical isolates was surprising. Two Lambda clinical isolates (Fig. 2b, Lambda (S1) and (S2)) with slightly different spike sequences were analyzed (Supplementary Table 1), and the results were consistent. The reporter SARS-CoV-2 system is powerful because the only difference between the viruses being analyzed is the spike, whereas natural isolates contain differences throughout the genome. It's possible that changes in other gene products (e.g., membrane protein or envelope protein) could impact neutralization phenotype, but this remains to be understood.

major Omicron subvariants that circulated in the US between December 2021 to March 2022 (Fig. 3c).

We further confirmed that the sharp increase of neutralizing titers from pre-booster to post-booster was a result of the booster vaccination rather than breakthrough infections. Using a Meso Scale Discovery (MSD) assay, we determined that antibodies specific to the SARS-CoV-2 nucleocapsid protein remained at very low levels (0.60–0.99 binding antibody units (BAU)/mL) in both pre-and post-booster sera (Fig. 4a). In contrast, the receptor binding domain (RBD)-specific antibodies and spike-specific antibodies increased 50-fold from 174 BAU/mL to 8636 BAU/mL and 41-fold from 147 BAU/mL to 5976 BAU/mL, respectively (Fig. 4a). This level of increase was comparable to the 51-fold increase of neutralizing titers from pre-booster to post-booster sera (Fig. 3a, b). When a new MSD kit containing spike proteins from VOCs/VOIs was used, 32–46-fold increases of antibody units was detected in pre-booster sera versus in post-booster sera, depending on the specific variant spike (Fig. 4b). However, compared to the fold-reduction (8.5–16) in the neutralization assay (Fig. 3), the fold-reduction of Omicron relative to 614D was smaller in the MSD assay (3.6–7.8) (Fig. 4b), probably due to the presence of other binding but non-neutralizing antibodies in the sera. Other MSD kits or non-MSD technologies could be explored if a surrogate for live virus neutralization assay is desired.

**Inhibition of virus replication by vaccinee sera.** To better understand the effect of variant spike genes on viral fitness in the presence and absence of neutralizing antibodies, we compared the replication of reporter viruses with the variant spikes in Calu-3 cells, a human lung epithelial cell line (Fig. 4). In the absence of post-second dose sera, many variants, including Delta and Omicron, replicated to comparable titers, but a few variants replicated to much higher or lower titers indicating the spike mutations can affect viral fitness in Calu-3 cells. Nevertheless, by comparing the viral titers in the presence or absence of sera, the inhibitory effect of the neutralizing sera on each variant can be quantified separately regardless of the titer

variation in the absence of sera. In the presence of post-second dose sera, even highly diluted, the titers of 614D and 614G viruses were reduced by more than 7-fold at 2X sera concentration (concentration of sera to achieve $FRNT_{50} = 2$ against the 614D virus) and more than 300-fold at 5X sera concentration ($FRNT_{50} = 5$). As anticipated from neutralization escape data (Fig. 2a), the Omicron variant replicated efficiently in the presence of both concentrations of sera (Fig. 5a). The inhibitory effect of the sera on most variants was generally correlated with their susceptibility to neutralization though there were exceptions. For example, both the Gamma and Delta variants had similar susceptibility to neutralization (Fig. 2a, b), but the Delta variant replicated relatively efficiently and only had a 3-fold reduction in viral titer at 5X sera concentration whereas the Gamma variant had a 20-fold reduction at the same sera concentration (Fig. 5a). The Delta variant's ability to replicate efficiently in the presence of sub-neutralizing concentrations of antisera may have facilitated infections of people with low-to-modest levels of neutralizing antibodies induced by prior infection or vaccination.

As the 5X post-second dose sera did not show any inhibitory effect on the replication of the Omicron/BA.1 reporter virus (Fig. 5a), we increased the sera concentration to 10X and 20X and also included the Omicron/BA.2 (Fig. 5b). While the inhibition was not as dramatic as on the 614D virus, which reduced the viral titers by more than 1000-fold, the 10X and 20X sera showed a dose-dependent inhibition of the Omicron BA.1 and BA.2 subvariants. The 10X sera reduced the BA.1 and BA.2 viral titers by 4–6-fold and the 20X sera reduced those subvariants by 10–25-fold. Interestingly, when the same test was performed using post-third dose sera, the 10X sera (based on $FRNT_{50} = 10$ against the 614D virus) reduced the BA.1 and BA.2 viral titers by 50–122-fold and the 20X sera reduced those subvariants by 687–765-fold (Fig. 5c). These replication inhibition results are consistent with the neutralization results, in which the post-third dose sera were more broadly neutralizing than the post-second dose sera, as the former displayed a 9–10-fold reduction in neutralizing titers against the BA.1 and BA.2 subvariants

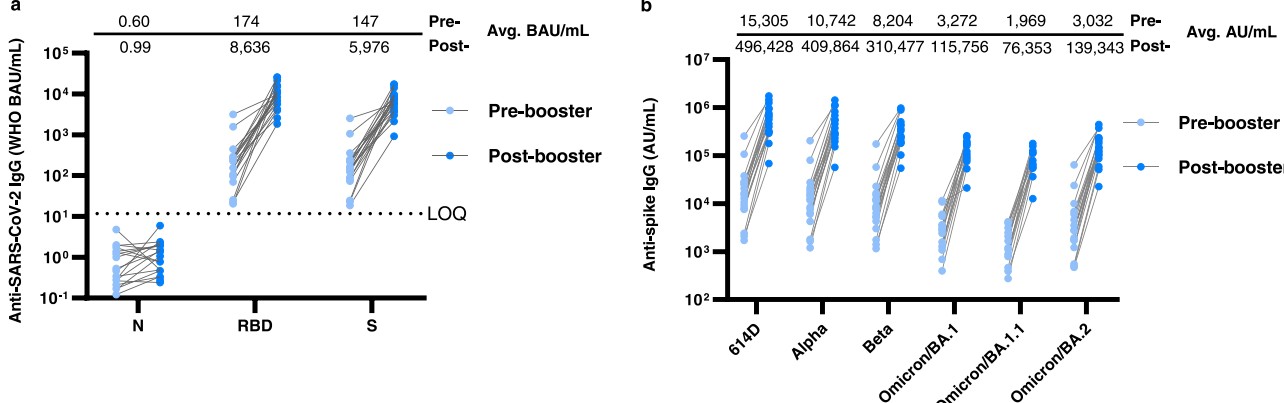

**Fig. 4 Evaluation of anti-SARS-CoV-2 IgG binding activity in mRNA vaccinee sera (pre- and post-booster).** Each dot represents calculated IgG concentration (BAU/mL or AU/mL) of an individual serum sample against a specific protein antigen. The geometric mean IgG concentrations (avg. BAU/mL or AU/mL) from pre- and post-booster sera are shown on the top of the graph. For statistical analysis, a two-tailed Wilcoxon matched-pairs signed-rank test was performed by comparing antibody concentration of pre- and post-booster sera. Test statistics and *P* value are summarized in Supplementary Table 2. Source data are provided as a Source Data file. **a** IgG antibodies specific to SARS-CoV-2 nucleocapsid (N), receptor binding domain (RBD) and spike(S) were measured. $N = 18–20$ biologically independent pre-booster sera (light blue) and $N = 18–20$ biologically independent post-booster sera (dark blue). The average IgG concentrations in all post-booster sera differ significantly ($P < 0.0001$) from the pre-booster sera, except those specific to the nucleocapsid ($P = 0.1084$). Dashed line represents the limit of quantification for the N protein. LOQ = 11.8. **b** IgG antibodies specific to SARS-CoV-2 spike proteins of 6 different viruses were measured. $N = 17–20$ biologically independent pre-booster sera (light blue) and $N = 17–20$ biologically independent post-booster sera (dark blue). The average IgG concentrations in all post-booster sera differ significantly ($P < 0.0001$) from the pre-booster sera. BAU: binding antibody units calibrated to WHO international standard. AU arbitrary units.

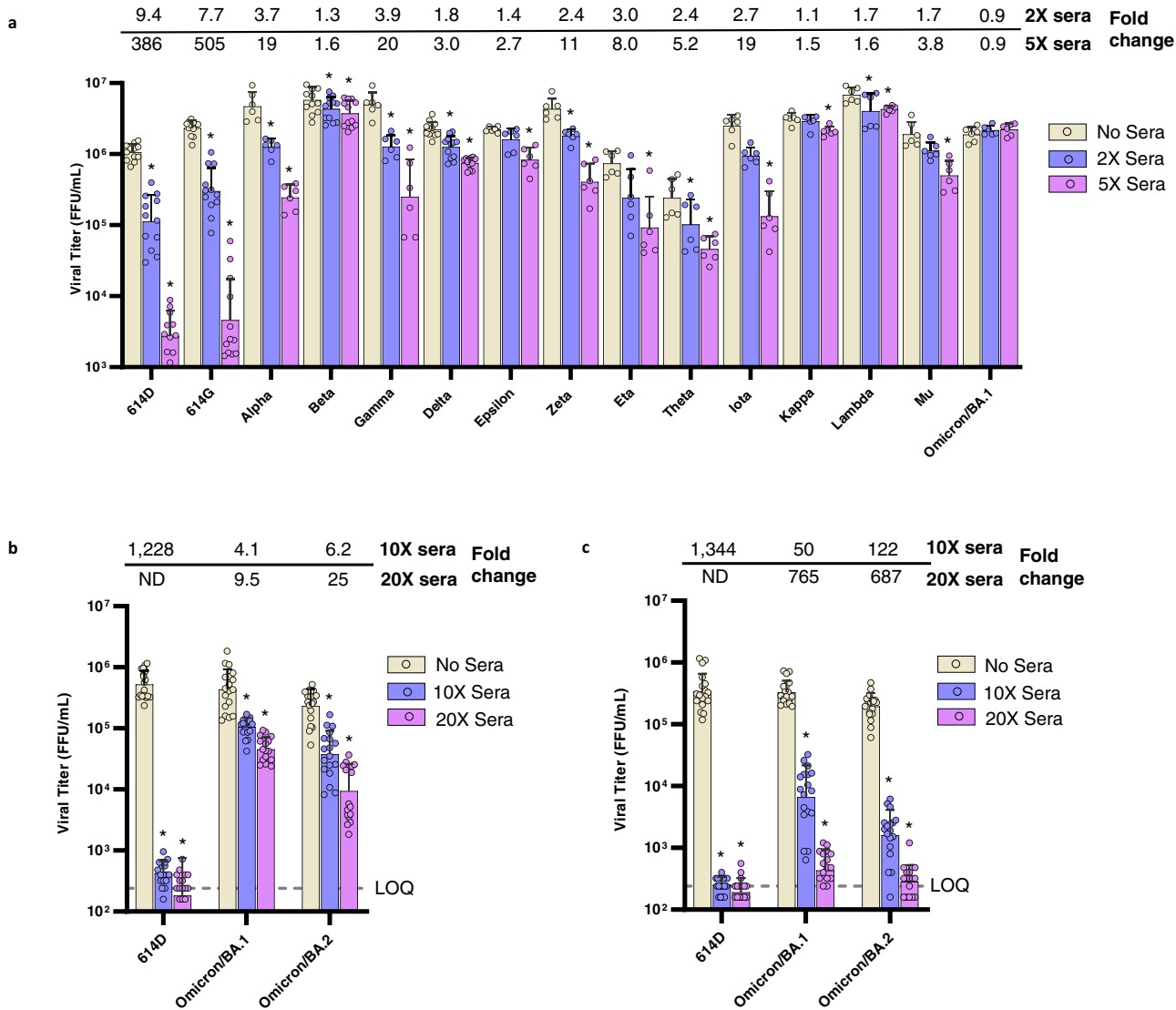

**Fig. 5 Inhibition of mRNA vaccinee sera against live SARS-CoV-2 viruses.** Calu-3 cells were infected with 200–400 focus forming units (FFU) of each virus and incubated for 2 days in media with or without sera. The viruses were collected from Calu-3 supernatant at 2 days post inoculation and titrated by FFU assay. The geometric mean viral titers of variants under different treatment conditions are represented in the graph as bars with geometric standard deviation. Bar graphs present the titers of each variant under different treatment conditions. Error bars representing titer differences are marked as *, representing $p < 0.05$, compared to the no sera control within each virus group. Two-tailed Wilcoxon matched-pairs signed-rank test was used for statistical significance analysis and the statistics and $P$ value are summarized in Supplementary Table 2. Source data are provided as a Source Data file. Titer fold changes (reductions) compared to the no sera control are shown on the top of the panel. **a** The sera for incubation were pooled from the individual sera used in Fig. 2a and diluted to 2X or 5X concentration (diluted sera titer $FRNT_{50} = 2$ or 5 against 614D reference virus). $N = 6$ or 12 viral titers over 2 biologically independent sera pools in 1–2 independent experiments. **b** The sera for incubation were pooled from the individual sera used in Fig. 2a (post-second dose) and diluted to 10X or 20X concentration (diluted sera titer $FRNT_{50} = 10$ or 20 against 614D reference virus). $N = 18$ viral titers over 2 biologically independent sera pools in 1–2 independent experiments. The fold change for the 614D-20X sera group is not determined (ND) as the average viral titer of that group is below the LOQ. **c** The sera for incubation were pooled from the individual sera used in Fig. 3b (post-third dose) and diluted to 10X or 20X concentration (diluted sera titer $FRNT_{50} = 10$ or 20 against 614D reference virus). $N = 18$ viral titers over 2 biologically independent sera pools in 1–2 independent experiments. The fold change for the 614D-20X sera group is not determined (ND) as the average viral titer of that group is below the LOQ. Yellow bars, viral titers in the absence of sera; blue bars, viral titers in 2X or 10X sera; purple bars, viral titers in 5X or 20X sera. Limit of quantification LOQ = 240 FFU/mL.

(relative to 614D) and the latter displayed a 25–31-fold reduction (Figs. 2 and 3).

## Discussion

Using serum samples collected from US volunteers 2–6 weeks and 6–7 months after receiving the second dose of COVID-19 mRNA vaccine and 2–6 weeks after receiving the third dose of mRNA vaccine, we systematically characterized at least one representative virus from each of the 13 past and current VOCs and VOIs designated by WHO, from the earliest Alpha variant to the recent Omicron subvariants. We rapidly generated many of the recombinant SARS-CoV-2 reporter viruses and determined post-vaccination sera neutralizing titers with these viruses using a

FRNT assay, which was confirmed to produce consistent results in a large-scale comparative neutralization assay study involving 12 independent laboratories[15]. We also isolated viruses from clinical specimens, including most of the VOCs and VOIs, shared these viruses with many external partners, and tested them in FRNT assays. The FRNT results from the reporter viruses and the clinical isolates were almost always comparable with few exceptions. Both sets of data clearly showed that different variants had differential resistance to neutralization by the vaccinee sera, with the Omicron variant being the most resistant.

While most post-second-dose sera (2–6 weeks after second dose of vaccine) neutralized the pre-Omicron variants with $FRNT_{50}$ titers higher than 40, many of them dropped titers strikingly to between 5 and 40 against the Omicron variant (Fig. 2). More concerning were the sera collected 6–7 months after the second dose of vaccine (pre-booster sera), as half of those had titers below 5 $FRNT_{50}$ (Fig. 3a). It has been widely accepted in the influenza vaccine field that a neutralizing titer (or hemagglutination inhibition titer) of 40 or higher is deemed protective (>50% reduction of infection rate)[16] and an 8-fold reduction in neutralization in vitro is consideration for updating the influenza vaccine strain composition. For COVID-19 vaccines, the few studies on correlates of protection that have been published suggest a titer of 50–100 is the minimum protective neutralizing titer[17,18], while the fold reduction warranting a vaccine antigen update has yet to be determined. This is also complicated by the unknown role of other effectors in vaccine protection such as memory B cell or T cell immunity[19]. However, the level of neutralizing antibody titer is likely predictive of the level of immune protection[20–23]. As neutralizing antibodies wane over time[24] (Figs. 2a and 3a), infections in fully vaccinated persons by variants circulating at high prevalence are likely to increase. This is especially true for the Omicron variant that has a high ability to escape neutralization[25]. Nevertheless, vaccines do prevent and attenuate COVID-19[26] and anamnestic responses provided through rapid expansion of memory cells should accelerate viral clearance. Most importantly, a third dose of vaccine (post-booster sera) did increase the neutralizing titer against Omicron subvariants to approximately 10-fold higher than that of the post-second dose sera and 80-fold higher than that of the pre-booster sera (Figs. 2 and 3). Therefore, closely monitoring the emergence of variants resistant to neutralization is a necessary and urgent task, and vaccination, including booster doses, remains the most effective strategy to combat the COVID-19 pandemic.

## Methods

**Ethics statement**. Vaccinee serum samples were collected from individuals through the Influenza and Other Viruses in the Acutely Ill (IVY) Network, a Centers for Disease Control and Prevention (CDC)-funded collaboration to monitor the effectiveness of SARS-CoV-2 vaccines among US adults. Participants had no prior or current diagnosis of infection with SARS-CoV-2 and were fully vaccinated (at least 14 days after the second dose or the third dose) with either Pfizer-BioNTech mRNA vaccine BNT162b2 or Moderna mRNA-1273 vaccine. This activity was approved by each participating institution, either as a research project with written informed consent or as a public health surveillance project without written informed consent. This activity was also reviewed by the CDC and conducted in a manner consistent with applicable federal laws and CDC policies: see e.g., 45 C.F.R. part 46.102(l)(2), 21 C.F.R. part 56; 42 U.S.C. §241(d); 5 U.S.C. §552a; 44 U.S.C. §3501 et seq.

**Biosafety statement**. All work involving infectious SARS-CoV-2 virus, including recombinant reporter virus, was performed in CDC Biosafety Level 3 facilities with enhanced practices (BSL-3E). All personnel working with the virus were trained with relevant safety and procedure-specific protocols and their competency for performing the work in the BSL-3E laboratories was certified. Recombinant DNA work was approved by CDC's Institutional Biosafety Committee (IBC). For sequencing, virus was inactivated following protocols approved by CDC's Laboratory Safety Review Board (LSRB) with a witness confirming that all steps were performed correctly to ensure complete inactivation of virus. After receiving appropriate approvals, inactivated virus was transferred to BSL-2E laboratories for downstream processing.

**Prevalence analysis of variants**. SARS-CoV-2 variant statistics for US national genomic surveillance specimens—reported to NCBI and GISAID by the National SARS-CoV-2 Strain Surveillance (NS3) network, CDC-contracted diagnostic and research laboratories, and US baseline surveillance program—were compiled and rendered in Tableau Desktop (version 2021.1.1). Daily proportionalities were aggregated by attributed Pangolin (version 4.0.6) lineage assignment including variants of concern (VOC), variants of interest (VOI), and lineages with published World Health Organization (WHO) nomenclature[7]. Pangolin sub-lineages with shared WHO aliases were consolidated: B.1.1.7 and Q sub-lineages (Alpha); B.1.351, B.1.351.2, B.1.351.3 and B.1.351.5 (Beta); P.1 and P.1 sub-lineages (Gamma); B.1.617.2 and AY sub-lineages (Delta); B.1.621 and B.1.621.1 (Mu); and B.1.1.529 and BA.1/BA.2 sub-lineages (Omicron). Unassigned variants and Pangolin lineages encoding an aspartate (D) or glycine (G) at position 614 were assigned respective "614D" and "614G" labels. Variants which did not satisfy the above criteria were consolidated into "Other Lineage(s)." Clinical statistics included all confirmed and probable daily cases and deaths reported to CDC surveillance network (last updated on May 31, 2022).

### Generation of SARS-CoV-2 reporter viruses

*Risk-benefit analysis.* A comprehensive risk-benefit analysis was conducted for using recombinant SARS-CoV-2 reporter viruses in neutralization assays. Briefly, the benefits of using the reporter viruses are: (1) enabling rapid characterization of variants before they are detected in the United States or before CDC receives specimens; (2) eliminating all fixation and staining steps in neutralization assays, shortening the time infectious samples are handled, and reducing chemical safety risks (e.g., formalin) by removing the need to fix cells; (3) minimizing the impact of substitutions in non-spike genes on neutralizing titers, as changes solely reflect the effect of spike mutations; (4) enabling assessment of impact of individual or specific sets of spike mutations; (5) enabling more consistent comparisons as isolates from different clinical specimens were noted to have distinct growth properties even though they were from the same lineage. The associated risk assessments are: (1) reporter viruses are different from any natural virus and created by introducing the spike mutations from a new variant into the backbone virus (progenitor strain Wuhan-Hu-1). The transmissibility of a particular resultant virus could be somewhere between the progenitor virus and the natural variant; (2) as there is limited epidemiological or clinical evidence to suggest spike mutations present in SARS-CoV-2 variants increase pathogenicity, it is most likely the pathogenicity of the reporter viruses will be equivalent or reduced as compared to the progenitor strain or the variant strain; (3) all naturally occurring SARS-CoV-2 variants descending from the progenitor strain have acquired mutations in other genes along with the spike gene. It is possible that some of the non-spike mutations may decrease the transmissibility or pathogenicity of the variant, in which case a reporter virus may be more transmissible or pathogenic than the variant. However, sequence analysis and literature review indicate this risk is very low, especially regarding its potential public health impact during this ongoing pandemic. The safeguard and mitigation strategies are: (1) the backbone of the reporter virus is based on the Wuhan-Hu-1 strain, which is expected to be the least transmissible strain compared to later variants; (2) a *mNeonGreen* reporter gene replaces the *ORF7a* in the reporter virus, which may attenuate the virus as the ORF7a protein has been reported to be an interferon antagonist[27,28]; (3) mutations engineered into a reporter virus are either part of or all of the spike mutations found in a natural isolate and the engineering of unnatural mutations is prohibited; (4) the reporter viruses are only to be used in in vitro studies, such as neutralization assays, and not in in vivo studies; (5) all the in vitro work is conducted in BSL-3E facilities including enhanced practices such as shower out after experiments to minimize the possibility of accidental release of the reporter virus to the environment; (6) all staff working with the reporter viruses are fully vaccinated; (7) all staff are approved for working with BSL-3E select agents with senior staff having decades of BSL-3E experience working with highly pathogenic viruses. The conclusion is: under the current public health emergency, with the urgency for antigenic surveillance of variants, the benefits of using SARS-CoV-2 reporter viruses exceeds the risks associated with generating and using recombinant reporter viruses. These risks are believed to be extremely low after mitigation.

*DNA construct.* The DNA constructs were either generated as linear fragments as previously described[29] or generated as a cloned DNA as detailed here. The DNA clone for SARS-CoV-2 strain Wuhan-Hu-1 (GenBank accession number: NC_045512) was purchased from Codex DNA (San Diego, CA, SC2-FLSG-1111). The viral genome was flanked by a T7 promoter sequence at the 5′ end and a linearization site at the 3′ end. The whole cassette was cloned into a bacterial artificial chromosome (BAC) vector. The DNA clone was modified to replace the *ORF7a* gene with a human codon-optimized *mNeonGreen* gene (GenBank accession number: AGG56535.1) following the same design as reported previously[29]. The spike gene of this progenitor reporter virus was excised by AscI and BamHI-HF restriction enzymes, resulting in a linearized vector into which synthetic variant spike genes can be assembled using Gibson Assembly (NEB). The Gibson

Assembly reaction was then transformed into TransforMax™ EPI300™ Electrocompetent *E. coli* (Lucigen, EC300150). Transformations were immediately recovered in SOC medium at 30 °C for 1 h, and plated on LB agar plates containing 25 µg/ml chloramphenicol, followed by approximately 2 days of incubation at 30 °C. Colonies were picked and inoculated into LB broth containing 25 µg/ml chloramphenicol for approximately $16 \pm 2$ h followed by induction for approximately $4 \pm 1$ h at 30 °C. DNA was extracted and the sequence was verified by Illumina next-generation sequencing (NGS).

*In vitro transcription.* Infectious clones were linearized by SbfI-HF digestion and cleaned up by phenol:chloroform:isoamyl alcohol (PCIA) (25:24:1) extraction. Full-length viral RNA was generated using the T7 RiboMAX™ Express Large Scale RNA Production System with slight modifications to manufacturer's instructions (Promega, P1320). Briefly, reaction components were adjusted such that in a 50 µL reaction the final concentration of ATP, CTP, and UTP was 7.5 mM, GTP 3.5 mM, and the Anti-Reverse Cap Analog (NEB, S1411S) was used at 2.8 mM. After 2–3 h of incubation at 30 °C, RNA was cleaned up by PCIA and ethanol precipitated for at least 1 h. Quality of the RNA was assessed by UV-vis spectroscopy and denaturing agarose gel electrophoresis.

*Nucleocapsid protein expressing cell line.* Vero E6 cells (ATCC, CRL-1586) were transfected using Lipofectamine 3000 (Invitrogen, L3000001) with a plasmid encoding SARS-CoV-2 nucleocapsid protein via CMV3 promoter as well as mCherry2 via an IRES element. Transfected cells were placed under drug selection (0.1–0.3 mg/ml geneticin) to establish the pooled bulk population. Stable single-cell clones were selected from the bulk population by serial dilution plating and drug selection. The expression of nucleocapsid protein was confirmed by the SARS-CoV-2 Nucleocapsid Protein ELISA Kit (ABclonal, RK04136) and the cell clone supporting the most efficient virus rescue was selected (VeroE6-N). Cells were maintained in DMEM supplemented with 10% FBS and 0.2 mg/ml geneticin.

*Virus rescue.* To rescue the SARS-CoV-2 reporter virus, VeroE6-N cells were trypsinized, washed with Opti-MEM (ThermoFisher, 31985062) and resuspended in 100 µL nucleofector solution at a concentration of $1.5 \times 10^6$ cells/100 µl following the instructions of the Nucleofector Kit V (Lonza, VCA-1003). In vitro transcribed RNA (5 µg) was added to the cells and the cell-RNA mixture was transferred into an electroporation cuvette. Electroporation was completed using the Program T-024 of the Nucleofector 2b device (Lonza, AAB-1001). Electroporated cells were immediately transferred into a 6-well plate pre-filled with 2 ml/well of pre-warmed Opti-MEM. At 18–24 h post-transfection, supernatant was collected (P0) and inoculated onto a monolayer of VeroE6/TMPRSS2 cells[14] (JCRB Cell Bank, JCRB1819). Twenty-four hours post-inoculation, supernatant was collected to make the seed stock (P1). P1 was propagated in T-150 flasks of VeroE6/TMPRSS2 cells at a multiplicity of infection (MOI) of 0.02–0.1 for 24 h to make the P2 working stock. The working stock was sequenced as described below.

*Sequence confirmation.* All the SARS-CoV-2 reporter viruses were sequenced by NGS to confirm the sequence of the spike gene. Total RNA was extracted from the working stock of each reporter virus and treated with DNase using the DNase Max kit (Qiagen, 15200-50) following manufacturer's instructions. Five microliters of resulting clean RNA were used for first- and second-strand cDNA synthesis and library preparation using NEB Ultra II Directional RNA library prep kit for Illumina (New England Biolabs, E7760S/L). Libraries were barcoded with unique dual indices synthesized in the CDC Biotechnology Core Facility Oligonucleotide Synthesis Laboratory. Resulting libraries were analyzed for size using the Agilent Fragment Analyzer (Agilent Technologies, Inc., Santa Clara, CA) and quantified using the Qubit 4 Fluorometer (Thermo Fisher Scientific, Waltham, MA). Libraries were normalized to equimolar concentrations, pooled, and sequenced on Illumina NovaSeq 6000 (Illumina, San Diego, CA, USA) using the NovaSeq v1.5 SP Reagent Kit (300 cycles). Demultiplexed reads were processed and assembled using the Iterative Refinement Meta-Assembler (IRMA) on a custom CoV-recombinant configuration[30]. The 614D reporter virus (Wuhan-Hu-1 strain with the *ORF7a* gene replaced by *mNeonGreen*) was used as the reference. Reads were filtered for a minimum median phred score (Q score) of 27 and a minimum read length of 80 bases. A Striped Smith-Waterman algorithm was selected for read alignment, and final assembly was performed against the reference sequence matched during read gathering. Amended consensus genomes were created from plurality assemblies by ambiguation of bases with coverage < 20x to "N", and positions with a minor allele frequency (MAF) > 0.2 were given ambiguous nucleotide codes according to IUPAC conventions. Quality metrics were calculated using a count of non-ambiguated amended consensus bases to show the proportion of recombinant genome assembled, and average coverage depth across the genome was noted. The full genome sequences of all the viruses are being deposited in GenBank with accession numbers ON571504 – ON571519.

**Meso scale discovery (MSD) immunoassay.** Serum samples were analyzed at 1:5000 and 1:50,000 dilutions for IgG to SARS-CoV-2 nucleocapsid (N), SARS-CoV-2 S1 receptor binding domain (RBD), and SARS-CoV-2 spike (S) protein using the V-PLEX SARS-CoV-2 Panel 2 (IgG) Kit (Meso Scale Discovery,

K15383U-2), as described previously[31]. Serum antibody levels were calculated using Reference Standard 1 and converted to WHO International Binding Antibody Units (BAU/mL) per manufacturer kit instructions. The serum samples were also analyzed at 1:5000 and 1:50,000 dilutions for IgG to spike proteins of different SARS-CoV-2 variants using the V-PLEX SARS-CoV-2 Panel 25 (IgG) Kit. However, as calibration of the variant spikes with the WHO International Standard had not been completed by the manufacturer, the data were presented in Arbitrary Units (AU/mL) instead of BAU/mL.

**Focus reduction neutralization Test (FRNT)**

*Reporter virus-based assay.* Serum specimens were heat-inactivated at 56 °C for 30 min, aliquoted, and stored at −80 °C. Each serum sample was serially diluted in 3-fold steps for 7 dilutions (starting at 1:5, 1:10, or 1:40) in sextuplicate in 96-well round bottom plates. SARS-CoV-2 reporter virus was diluted to 3200–4000 focus forming units (FFUs) per ml. Diluted serum samples were mixed with an equal volume of diluted virus and incubated for 1 h at room temperature (21 ± 2 °C). Media from confluent monolayer VeroE6/TMPRSS2 in 96-well tissue culture plates was removed, and 50 µl of the serum–virus mixture was inoculated into each well of cells and incubated at 37 °C in a 5% $CO_2$ atmosphere for 2 h. The wells were overlaid with 100 µl of 0.75% methylcellulose in DMEM (Gibco, 11-965-092), supplemented with 2% HI-FBS and 1x Pen-Strep and incubated at 33 °C in a 5% $CO_2$ incubator for 16–18 h. Plates were scanned using a CellInsight CX5 High-Content Screening Platform (Thermo Scientific) running an "Acquisition Only" protocol within Cellomics Scan Version 6.6.0 (Thermo Scientific, Build 8153). All plates were imaged under equal exposure conditions per channel and under 4x magnification.

Foci were identified and quantified using appropriate "Spot Detection" protocol within Cellomics Scan Version 6.6.2 (Thermo Scientific, Build 8533). Spot counts for each channel were exported for further analysis in R (Version 4.0.3). $FRNT_{50}$ values were calculated by fitting the three-parameter log-logistic function (LL.3) to the FFU counts paired with corresponding dilution information. In cases where the Hill Constant was fit at less than 0.5, e.g., incomplete neutralization, $FRNT_{50}$ values were estimated with a two-parameter fit while fixing the Hill Constant to 1. The R script has been deposited in GitHub: https://github.com/CDCgov/SARS-CoV-2_FRNTcalculations/.

*Clinical isolate-based assay.* SARS-CoV-2 isolates were propagated on Vero/TMPRSS2 cells. All stocks were inoculated at multiplicity of infection (MOI) of approximately 0.004 or 0.01 (for Omicron viruses) and harvested at 2 days post-inoculation. The viral spike sequences were verified via unbiased NGS sequencing (KAPA HyperPrep library kit with RiboErase, followed by Illumina sequencing). All cells and virus stocks tested negative for mycoplasma using MycoAlert Plus reagents (Lonza, LT07-703). Heat-inactivated serum samples were serially diluted in 3-fold steps in DMEM supplemented with 2% heat-inactivated fetal bovine serum (HI-FBS), 1x Pen-Strep and sodium pyruvate (Gibco). The serum dilutions were mixed with an equal volume of virus in the same medium (final serum dilutions 1:10–1:7,290) and incubated for 1 h at 37 °C. Vero/TMPRSS2 cells growing in 96-well imaging plates were then inoculated in triplicate with 40 µL of serum-virus mixtures and incubated for 1 h at 37 °C with periodic shaking of the plates. Inocula were removed and cells overlaid with 1.5% medium viscosity carboxymethylcellulose (Sigma-Aldrich, 9004-32-4) in MEM (Gibco, 11095080), supplemented with 4% HI-FBS, 1x Penicillin-Streptomycin (pen/strep), and sodium pyruvate. Twenty hours later, the overlay was washed off with PBS, and cells were fixed with 10% neutral-buffered formalin, permeabilized with 0.5% Triton X-100 in PBS, blocked with 1% bovine serum albumin in PBS, and stained using SARS/SARS-CoV-2 Coronavirus Nucleocapsid Monoclonal Antibody (Invitrogen, MA5-29981) as the primary antibody at 1:4000 dilution followed by Alexa647-conjugated secondary antibody (Invitrogen, A32728) at 1:400 dilution. The monolayers were imaged using a BioTek Cytation3 instrument and virus foci (approximately 100–200/well in no-serum control wells) were counted using Gen5 software. The foci counts were normalized to no-serum controls, and 4 parameter nonlinear regression analysis with bottom constraint set to 0, and top value set to 1 (GraphPad Prism v7.04) was used to fit a curve to the data and to determine the $FRNT_{50}$ value.

**Virus replication in Calu-3 cells.** Calu-3 cells (Human lung epithelial cell line) were obtained from CDC's Division of Scientific Resources (DSR) (ATCC, HTB-55). Cells were seeded in 12-well plates and cultured 4–5 days until cell confluence reached 80–90% before infection. Culture media was removed from the cells before infection and 200–400 focus forming units (FFU) of virus was added into each well (triplicate wells for each virus). The plates were incubated at 37 °C in a 5% $CO_2$ atmosphere for 1 h. Ten individual vaccinee serum samples from persons who received Pfizer-BioNTech mRNA vaccine BNT162b2 and 10 vaccinee serum samples from persons who received Moderna mRNA-1273 vaccine were each normalized to 500 $FRNT_{50}$ (against 614D reference virus) and pooled separately (500X stock). Pooled Moderna or Pfizer sera was diluted to 2X, 5X, 10X or 20X concentration (concentration to achieve $FRNT_{50}$ = 2, 5, 10, or 20 against the 614D reference virus) in infection media (DMEM supplemented with 2% HI-FBS and 1x pen/strep). The inoculum was removed from each well after incubation and 1 ml of infection media with or without the diluted sera was added to corresponding wells (0X, 2X, 5X, 10X,

and 20X $FRNT_{50}$) and the plates were returned to the 37 °C, 5% $CO_2$ incubator for further incubation. Two days later, the culture supernatant was collected and titrated by FFU assay. The FFU assay was performed similarly to the FRNT assay by serial dilution of the virus and without mixing the virus with any sera. The foci acquisition and quantification steps were the same as described in the FRNT assay. For each variant, the viral titers in the presence of sera were compared to those in the absence of sera to calculate the fold of change (reduction) in titers.

*Data processing and statistical analysis.* Geometric mean titers (GMTs) of each virus were calculated using the $FRNT_{50}$ neutralizing titers of all the serum samples tested against that virus. For each variant, the average fold change is the geometric mean of the individual FRNT50 ratios (614D/variant) calculated for each serum sample. At least 20 serum samples were used in FRNT assay of each virus, including at least 10 sera from recipients of BNT162b2 and 9 sera from recipients of mRNA-1273. Graphs were made by GraphPad Prism (Version 8.4.2). For statistical analysis, a two-tailed Wilcoxon matched-pairs signed-rank test was performed to compare the neutralization titer of each variant with the reference virus for the same sera, or the virus titer in diluted sera against no sera using GraphPad Prism 9.3.1 and R version 4.1.2, with $P < 0.05$ considered significant. The number of samples, test statistics (W), Z value, effect size (r) and P value are summarized for each comparison in Supplementary Table 2. The effect size was calculated as previously described[32].

**Reporting summary**. Further information on research design is available in the Nature Research Reporting Summary linked to this article.

## Data availability

The full genome sequences of viruses generated in this study have been deposited in the GenBank database under accession code ON571504, ON571505, ON571506, ON571507, ON571508, ON571509, ON571510, ON571511, ON571512, ON571513, ON571514, ON571515, ON571516, ON571517, ON571518, ON571519. All other data generated in this study are provided in the Supplementary Information/Source Data file. Source data are provided with this paper. In addition, unique biological materials such as viruses can be requested from CDC with necessary documents such as MTA. Request for post-vaccination sera will be determined on a case-by-case basis due to IRB restrictions as well as limited remaining volume. Source data are provided with this paper.

## Code availability

The R script used to calculate the $FRNT_{50}$ titers has been deposited in GitHub: https://github.com/CDCgov/SARS-CoV-2_FRNTcalculations/ [https://doi.org/10.5281/zenodo.6639445].

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

## Acknowledgements

We acknowledge the hundreds of publications each containing neutralization results on some of the VOCs or VOIs reported in this manuscript for which we regret for not being able to cite due to the limitation on the number of references. We thank César G Albariño, Ginger Atteberry, Dennis Bagarozzi, Jessica Chen, Jennifer Folster, Matthew Keller, Jimma Liddell, Ji Liu, Gillian McAllister, Magdalena Medrzycki, Krista Queen, Shannon Rogers, Jarad Schiffer, Maria Solano, Sarah Talarico, Brett Whitaker, Jiangwei Yao, and Natosha Zanders for coordination, testing, or analysis support. This work was funded and supported by the CDC COVID-19 Emergency Response and part of the sequencing effort was made possible through support from the CDC Advanced Molecular Detection (AMD) program. Use of trade names is for identification only and does not imply endorsement by the US Centers for Disease Control and Prevention or the US Department of Health and Human Services. The findings and conclusions in this report are those of the authors and do not necessarily represent the official position of the US Centers for Disease Control and Prevention.

## Author contributions

D.E.W. and B.Z. conceived the study. C.T.D., C.F.S., N.J.T., and B.Z. designed the experiments. L.W., M.H.K, N.J., C.T.D., D.E.W., and B.Z. wrote the paper. L.W., M.H.K, N.J., H.D., G.B., L.M., M.C., P.S., B.M.C., M.S., B.R.M., J.H., X.L., S.L., E.A.P., J.J., D.C., P.C., M.H.J., E.K.M., G.P.L., M.H., J.L.H., A.T., Y.L., Y.T., K.Z., K.L., A.B., W.W., M.W., T.W., S.H.P., S.T., J.R.B., A.L.H., L.K.M., J.S.L., H.X., X.X., P.-Y.S., C.T.D., C.F.S., N.J.T., and B.Z. performed the experiments and/or analyzed the data. M.W.T., W.H.S., N.I.S., M.C.E., D.C.F., K.W.G., D.N.H., and M.P. coordinated or supervised the collection of serum samples from vaccinee volunteers. SSEV Bioinformatics Working Group tracked

and analyzed the prevalence of SARS-CoV-2 variants. This study was supervised by M.S.O., V.G.D., and D.E.W. as part of the SARS-CoV-2 variants characterization efforts of the Strain Surveillance and Emerging Variant Team within the Laboratory and Testing Task Force of the CDC COVID-19 Emergency Response. All authors reviewed and approved the manuscript.

## Competing interests
X.X. and P.-Y.S. have filed a patent on the reverse genetic system. X.X., and P.-Y.S. received compensation from Pfizer for COVID-19 vaccine development. Other authors declare no competing interests.

## Additional information

## SSEV Bioinformatics Working Group

Dhwani Batra[1], Andrew S. Beck[1], Jason Caravas[1], Reina Chau[1], Roxana Cintron-Moret[1], Peter W. Cook[1], Jonathan Gerhart[1], Christopher A. Gulvik[1], Norman Hassell[1], Dakota Howard[1], Kristen Knipe[1], Rebecca J. Kondor[1], Nicholas A. Kovacs[1], Kristine Lacek[1], Brian R. Mann[1], Laura K. McMullan[1], Kara Moser[1], Roopa Reddy-Nagilla[1], Clinton R. Paden[1], Benjamin Rambo-Martin[1], Sandra Mathew[1], Matthew W. Schmerer[1], Samuel S. Shepard[1], Philip Shirk[1], Richard A. Stanton[1], Thomas J. Stark[1], Erisa Sula[1], Kendall Tymeckia[1], Yvette Unoarumhi[1], Voleti Subbalakshmi[1] & Xiao-yu Zheng[1]

10          NATURE COMMUNICATIONS | (2022)13:4350 | https://doi.org/10.1038/s41467-022-31929-6 | www.nature.com/naturecommunications
