## [Peer Review File · Nature Communications]

Differential neutralization and inhibition of SARS-CoV-2 variants by antibodies elicited by COVID-19 mRNA vaccinesReviewers' Comments:

Reviewer #1:

Remarks to the Author:

Wang et al. analysed the neutralization of a panel of variants of concerns and interests by vaccinee sera (2-6 weeks post-second dose). They used SARS-CoV-2 reporter viruses and clinical isolates in comparison. Compared to the reference virus (614D), the Beta (B.1.351), Theta (P.3), Kappa (B.1.617.1), Mu (B.1.621) and Omicron (B.1.1.529) variants showed the lowest neutralization titers. Omicron showed the greatest escape using SARS-CoV-2 reporter viruses. Reduction in neutralisation titers determined using the clinical isolates were quite similar in comparison with the titers determined using the reporter viruses for all variants, except for Lambda (C.37) which showed a lower reduction in neutralizing titers with the clinical isolate. The authors also confirmed the better ability of post-booster sera to enhance the neutralisation of Beta and Omicron compared to pre-booster sera. Interestingly, the authors evaluated the effects of specific substitutions using reporter viruses on neutralising activity of vaccinee sera (2-6 weeks post-second dose). They confirmed the importance of some substitutions like E484K or N-terminal domain mutations. Finally, they analysed the effect of variant spikes on the viral fitness with or without neutralising antibodies from a pool of vaccinee sera (2-6 weeks post-second dose). They found that the Beta and Omicron variants (reporter viruses) replicated efficiently in the presence of sera. The Delta variant (reporter virus) also replicated efficiently and only had a slight reduction in titers with the highest concentration of sera tested in this study.

The paper is globally interesting. Even though the first data of the paper do not provide very novel information, the panel of variants that the authors tested using SARS-CoV-2 reporter viruses was quite impressive. The comparison with clinical isolates is also very valuable. The second part of the paper is more novel. It is crucial to understand which substitutions and which mutated regions have the highest impact on the reduction in neutralising activity of vaccine-induced antibodies. It may help to anticipate immune escape of future variants. Finally, it is very informative to know how spike variants may impact on viral replication in combination with vaccine-induced antibodies. Please find below some comments to potentially strengthen the paper.

Major comments:

1) Could the authors show the correlations of FRNT50 titers between their SARS-CoV-2 reporter variants and clinical isolates ?

2) Fig 2c: A differential neutralisation of modified variants by vaccine-induced antibodies was observed in the study. Actually, the presence of mutations can decrease the binding affinity of antibodies but could also enhance the binding with ACE2. Did the authors observe that their modified variants had a difference in infectivity compared to the "classical variants" ? In addition, it would be very valuable to link neutralising data with antibody binding data to try to understand a little bit more the mechanisms underlying the changes in neutralisation activity. Did the authors have any antibody binding data with these modified variant spike antigens ?

3) It is known that boosters enhance the quantity of antibodies, which clearly impacts on the viral neutralization but they might potentially modify the quality of antibodies and their affinity to some regions. Did the authors try to compare the changes in neutralization of their modified variants using post-booster sera instead of post-2nd dose sera (Fig 2c) ?

4) Did the authors analyse the viral fitness of variants with neutralising antibodies from post-booster vaccinees (Fig 2d) ?

Reviewer #2:

Remarks to the Author:

The manuscript by Wang, et al. describes a comparison of neutralization and inhibition of SARS-CoV-2 Variants by post-mRNA-vaccine sera. The authors use statistical tools to track the prevalence of SARS-CoV-2 variants in the US, and then use this data, along with the VOI/VOC designations to identify variants to compare. They primarily use a reporter virus reverse genetics system where the variant spike protein is put on a Wuhan-Hu-1 reporter backbone. They proceed to confirm some of their findings using clinical isolates corresponding to the variants. From this, the authors are able to conclude that despite decreases in neutralization efficiency, the mRNA vaccine (especially after boost), is able to neutralize most variants to some extent, demonstrating the public health importance of vaccination and boosting for SARS-CoV-2. Overall, the manuscript is well written, methodologically sound, and of broad interest. However, I do have a few minor concerns about some of the statistical analyses and data presentation.

Minor recommendations:

1. When discussing the error bars on the graph in figure 2 (line 555), the authors suggest they represent the standard deviation, when they appear to represent the geometric standard deviation.
2. In lines 555-557, the authors say that all differences were statistically significant ($P < 0.0001$) except for 614G. However, variant B.1.1.7 (alpha) appears to have a very similar geometric mean and geometric standard deviation. Seeing the n-values for these two groups would help interpret this data.

REVIEWERS' COMMENTS

Reviewer #1 (Remarks to the Author):

Wang et al. analysed the neutralization of a panel of variants of concerns and interests by vaccinee sera (2-6 weeks post-second dose). They used SARS-CoV-2 reporter viruses and clinical isolates in comparison. Compared to the reference virus (614D), the Beta (B.1.351), Theta (P.3), Kappa (B.1.617.1), Mu (B.1.621) and Omicron (B.1.1.529) variants showed the lowest neutralization titers. Omicron showed the greatest escape using SARS-CoV-2 reporter viruses. Reduction in neutralisation titers determined using the clinical isolates were quite similar in comparison with the titers determined using the reporter viruses for all variants, except for Lambda (C.37) which showed a lower reduction in neutralizing titers with the clinical isolate. The authors also confirmed the better ability of post-booster sera to enhance the neutralisation of Beta and Omicron compared to pre-booster sera. Interestingly, the authors evaluated the effects of specific substitutions using reporter viruses on neutralising activity of vaccinee sera (2-6 weeks post-second dose). They confirmed the importance of some substitutions like E484K or N-terminal domain mutations. Finally, they analysed the effect of variant spikes on the viral fitness with or without neutralising antibodies from a pool of vaccinee sera (2-6 weeks post-second dose). They found that the Beta and Omicron variants (reporter viruses) replicated efficiently in the presence of sera. The Delta variant (reporter virus) also replicated efficiently and only had a slight reduction in titers with the highest concentration of sera tested in this study.

The paper is globally interesting. Even though the first data of the paper do not provide very novel information, the panel of variants that the authors tested using SARS-CoV-2 reporter viruses was quite impressive. The comparison with clinical isolates is also very valuable. The second part of the paper is more novel. It is crucial to understand which substitutions and which mutated regions have the highest impact on the reduction in neutralising activity of vaccine-induced antibodies. It may help to anticipate immune escape of future variants. Finally, it is very informative to know how spike variants may impact on viral replication in combination with vaccine-induced antibodies. Please find below some comments to potentially strengthen the paper.

Major comments:

1) Could the authors show the correlations of FRNT50 titers between their SARS-CoV-2 reporter variants and clinical isolates?

Response: We thank the reviewer for asking about the FRNT50 titers. In our original manuscript we only provided the titer fold changes of each variant compared to 614D as we were planning to provide the FRNT50 titers in the Source Data file. However, with the reviewer's question, we added the FRNT50 titers to all the figures of the revised manuscript and we were glad that the figures looked much better and presented more meaningful information. Much appreciated for this question. Based on the FRNT50 titers on the figures, it's very clear that the titers between the reporter variants and the clinical isolates are quite close. For example, using the same post-booster sera, the titers of 614D, BA.1, and BA.2 reporter viruses were 3154, 356, and 311 and the titers of the 614D, BA.1 and BA.2 clinical isolates were 3250, 367, and 333 (Fig. 3b vs 3c). The only obvious discrepancy is the Lambda variant in Fig. 2a and 2b,

which were discussed in the results section (lines 120-130).

2) Fig 2c: A differential neutralisation of modified variants by vaccine-induced antibodies was observed in the study. Actually, the presence of mutations can decrease the binding affinity of antibodies but could also enhance the binding with ACE2. Did the authors observe that their modified variants had a difference in infectivity compared to the “classical variants”? In addition, it would be very valuable to link neutralising data with antibody binding data to try to understand a little bit more the mechanisms underlying the changes in neutralisation activity. Did the authors have any antibody binding data with these modified variant spike antigens?

Response: We thank the reviewer’s interest in our modified variants (VOCs plus or minus specific mutations). As the focus of this study was on using different serum samples to test a large panel of SARS-CoV-2 variants covering all the WHO designated VOCs/VOIs from Alpha to Omicron, we performed additional experiments, added more data, and re-organized the manuscript to emphasize these two aspects of 1) comparing different VOCs/VOIs and 2) comparing different time series of sera. As a result, we decided to remove the original Fig. 2c from the manuscript as those data distract the focus and interrupt logic flow of the current manuscript. However, all the questions raised by the reviewer were very thoughtful and intriguing and we have at least partial data to each of questions, and we are happy to share with the reviewer with our thoughts on the modified variants.

1. Regarding ACE2 binding and infectivity:

We have done a significant amount of spike-ACE2 binding assays on the VOCs and VOIs and some were published previously (Nature 592 (7852), 122-127; Nature 602 (7896), 307-313). However, we found that a higher binding affinity doesn’t always result in a higher infectivity/replication in cell lines. For example, the Alpha variant has very high affinity to ACE2, but its infectivity is not any higher than the 614G virus in VeroE6/TMPRSS2 cells. For the modified variants, we didn’t notice any significant difference in infectivity in VeroE6/TMPRSS2 cells and they all grew to similar peak titers (within 1 log). But using a mouse cell line for example, we did see the B.1.351 (beta) variant without the N501Y mutation had much lower infectivity compared to the “classical” beta variant, which is consistent with results from other studies on the N501Y. It is obvious that different cell lines or primary cells have different susceptibility to different variants (infectivity). Therefore, to satisfactorily address the question on the receptor binding and infectivity, much more work needs to be done on those modified variants, which is out of the scope of this neutralization study.

2. Regarding neutralizing data vs. antibody binding data:

Again, thanks to all these great suggestions. We performed antibody-antigen binding assays using MSD on the pre-booster and post-booster sera against Omicron and some other variants. We added a new figure to the manuscript (Fig. 4) for that data. In general, the fold change in the antibody binding assay is smaller than that in the neutralizing assay, probably due to the fact that many non-neutralizing antibodies can bind to the variants and complicated the results. We didn’t do MSD assay on the modified variants mainly because commercial kits are not available from MSD. We can make corresponding recombinant proteins for the modified variants and customize the MSD kit to do the assay, but with the binding data in Fig. 4, we predict the

antibody binding for the modified variants will have the same trend as we observed for the “classical variants”, that the fold change will be smaller than that in the neutralization assay.

In summary, from scientific perspective, we totally appreciate and agree with the reviewer’s comments on the modified variants. With the shift of priorities at CDC for the COVID-19 response, we are not sure how much more efforts we can put on the modified variants. There are relevant studies have been published to investigate specific spike mutations using recombinant proteins or pseudotyped viruses. We are sure more exciting discoveries can be made by researchers from academia.

3) It is known that boosters enhance the quantity of antibodies, which clearly impacts on the viral neutralization but they might potentially modify the quality of antibodies and their affinity to some regions. Did the authors try to compare the changes in neutralization of their modified variants using post-booster sera instead of post-2nd dose sera (Fig 2c) ?

Response: We performed an extensive analysis of the post-second dose sera (2-6 weeks post-second dose), pre-booster sera (6-7 months post-second dose), and post-booster sera (2-6 weeks post third dose) for their neutralizing effects on the Omicron subvariants (new Fig. 2c, Fig. 3 and Fig 4). As the reviewer pointed out, the booster dramatically enhanced the quantity of antibodies and in our study, increased FRNT50 titer from around 800 to more than 3,000 for the 614D virus (Fig 2 and Fig 3). Moreover, compared to the post-second dose sera, the post-booster sera showed a smaller fold reduction of titers for Omicron relative to 614D (Fig. 2a vs. 3b and Fig. 2c vs. 3c). This result strongly indicates that the post-booster sera are likely to contain a higher ratio of cross-neutralizing (or broadly neutralizing) antibodies against the Omicron than the post-second dose sera and thus the booster modified the quality of the antibodies. Therefore, although we didn’t do additional comparison of the two types of sera using the modified variants, we predict the post-booster sera will neutralize some the modified variants better than the post-second dose sera (both relative to 614D).

4) Did the authors analyse the viral fitness of variants with neutralising antibodies from post-booster vaccinees (Fig 2d) ?

Response: We were also interested in this comparison and following the reviewer’s suggestion, we performed additional experiments on the Omicron BA.1 and BA.2 using higher concentrations of post-second dose sera and post-booster sera (post-third dose). The post-booster sera can more effectively inhibit the Omicron titers (reduce viral fitness) than the post-second dose sera, when they were applied at the same concentration (new Fig. 5b and 5c). These data are consistent with the neutralization data and further corroborate the notion that the booster modified the quality of the antibodies.

Reviewer #2 (Remarks to the Author):

The manuscript by Wang, et al. describes a comparison of neutralization and inhibition of SARS-CoV-2 Variants by post-mRNA-vaccine sera. The authors use statistical tools to track the prevalence of SARS-CoV-2 variants in the US, and then use this data, along with the VOI/VOC designations to identify variants to compare. They primarily use a reporter virus reverse genetics system where the variant spike protein is put on a Wuhan-Hu-1 reporter backbone. They proceed to confirm some of their findings using clinical isolates corresponding to the variants. From this, the authors are able to conclude that

despite decreases in neutralization efficiency, the mRNA vaccine (especially after boost), is able to neutralize most variants to some extent, demonstrating the public health importance of vaccination and boosting for SARS-CoV-2. Overall, the manuscript is well written, methodologically sound, and of broad interest. However, I do have a few minor concerns about some of the statistical analyses and data presentation.

Minor recommendations:

1. When discussing the error bars on the graph in figure 2 (line 555), the authors suggest they represent the standard deviation, when they appear to represent the geometric standard deviation.

Response: We thank the reviewer for pointing this out and we have now clarified in figure 2 and all other relevant figures that the error bars represent geometric standard deviation (line 644).

2. In lines 555-557, the authors say that all differences were statistically significant ($P < 0.0001$) except for 614G. However, variant B.1.1.7 (alpha) appears to have a very similar geometric mean and geometric standard deviation. Seeing the n-values for these two groups would help interpret this data.

Response: All statistics were originally done in SAS and we did not describe all the parameters clearly in our original manuscript. Thanks to the reviewer's question and the instructions from the Journal on statistics and on n-values. We have re-analyzed all data in this manuscript using the more commonly used two-tailed Wilcoxon matched-pairs signed-rank test. We compared the neutralization titer of each variant with the 614D virus for each serum sample and compared the virus titer in diluted sera against no sera with P value < 0.05 considered significant. The number of samples, test statistics (W), Z value, effect size (r) and P value are summarized for each comparison in Supplementary Table 2. We have described that in the methods section (lines 494-500) and in all figure legends. All the conclusions are still the same, including B.1.1.7 (Alpha) being significantly different from 614D in Fig. 2a, which may be a result of the powerful matched-pair analysis of the sera.